# Mechanism of Amyloid Gel Formation by Several Short Amyloidogenic Peptides

**DOI:** 10.3390/nano11113129

**Published:** 2021-11-20

**Authors:** Oxana V. Galzitskaya, Olga M. Selivanova, Elena Y. Gorbunova, Leila G. Mustaeva, Viacheslav N. Azev, Alexey K. Surin

**Affiliations:** 1Institute of Protein Research, Russian Academy of Sciences, 142290 Pushchino, Russia; seliv@vega.protres.ru (O.M.S.); alan@vega.protres.ru (A.K.S.); 2Institute of Theoretical and Experimental Biophysics, Russian Academy of Sciences, 142290 Pushchino, Russia; 3The Branch of the Institute of Bioorganic Chemistry, Russian Academy of Sciences, 142290 Pushchino, Russia; eyugorbunova@rambler.ru (E.Y.G.); mustaeva@rambler.ru (L.G.M.); viatcheslav.azev@bibch.ru (V.N.A.); 4State Research Center for Applied Microbiology and Biotechnology, 142279 Obolensk, Russia

**Keywords:** Aβ peptide, amyloid fibril, biofilm, oligomer, amyloidogenic regions

## Abstract

Under certain conditions, many proteins/peptides are capable of self-assembly into various supramolecular formations: fibrils, films, amyloid gels. Such formations can be associated with pathological phenomena, for example, with various neurodegenerative diseases in humans (Alzheimer’s, Parkinson’s and others), or perform various functions in the body, both in humans and in representatives of other domains of life. Recently, more and more data have appeared confirming the ability of many known and, probably, not yet studied proteins/peptides, to self-assemble into quaternary structures. Fibrils, biofilms and amyloid gels are promising objects for the developing field of research of nanobiotechnology. To develop methods for obtaining nanobiomaterials with desired properties, it is necessary to study the mechanism of such structure formation, as well as the influence of various factors on this process. In this work, we present the results of a study of the structure of biogels formed by four 10-membered amyloidogenic peptides: the VDSWNVLVAG peptide (AspNB) and its analogue VESWNVLVAG (GluNB), which are amyloidogenic fragments of the glucantransferase Bgl2p protein from a yeast cell wall, and amyloidogenic peptides Aβ(31–40), Aβ(33–42) from the Aβ(1–42) peptide. Based on the analysis of the data, we propose a possible mechanism for the formation of amyloid gels with these peptides.

## 1. Introduction

Fibrillar deposits are often associated with various diseases, including neurodegenerative ones (Parkinson’s, Alzheimer’s). To date, more than 30 different human pathologies are known to be associated with the detection of deposits in the body in the form of amyloid fibrils [1,2].

In addition to pathological amyloid structures, a large number of functional amyloids are present in the body. Such formations are inherent in many organisms and have been found in representatives of all domains of life [2]. It has been shown that a number of peptide/protein hormones are contained in secretory granules. To date, about 50 peptide mammalian hormones (glucagon, somatostatin, endorphin, calcitonin), which are stored in secretory granules in the form of fibrillar amyloid structures, are known. They are non-toxic and enter the body in monomeric form without loss of activity [3].

Many Enterobacteriaceae form curli, pili and tafi, which are components of the extracellular matrix, on the surface [2]. Such formations are fibrillar structures up to 1.5 μm in length and about 7–10 nm in diameter. The bacterial protein curli from *E. coli* is well studied [4]. Pili of some bacteria, for example, *Mycobacterium tuberculosis*, are involved in the formation of a common extracellular matrix with the host cells, which allows bacteria to survive in the host organism [5].

Functional amyloidogenic proteins of the bacterial cell wall can form, in addition to fibrillar structures on the cell surface, biofilms, which are involved in the formation of the extracellular matrix [2]. Both fibrillar formations and biofilms provide bacterial resistance in cell tissues of the host organism and lead to its colonization by pathogenic microorganisms. Due to the resistance of fibrillar formations to various external influences (temperature, pH, ionic conditions, proteases), biofilms also increase the resistance of microorganisms to drugs. Interestingly, biofilms promote bacterial adhesion to various non-biological surfaces: glass, metal, plastic and implants.

Amyloidogenic proteins have also been identified on the surface of some fungi. One class of such proteins (hydrophobins) is secreted on the cell surface and participates in the formation of fungal hyphae; it also assists hyphae to overcome the surface tension of water (water–air interface, hydrophobic–hydrophilic surface) [6].

The study of the amyloid formation process revealed that with an increase in the concentration of the studied preparations of small proteins and peptides, a gel-like precipitate is formed, i.e., the fibrils pass into a gel-like state. Gels have some properties of solids: they are able to maintain their shape and have strength, elasticity and plasticity. Initially, the properties of gels were studied using the example of polymers such as gelatin, pectin and various natural and synthetic polymers. These polymers were investigated by a method that was subsequently used to study deposits (fibril aggregates) in various tissues of humans and animals. To study the properties of gels from amyloidogenic proteins/peptides, the same approaches were used as for fibrils: EM, AFM, NMR, ThT binding, X-ray diffraction analysis and other research methods [6,7,8,9,10,11,12,13,14,15,16,17]. According to EM and AFM microscopy data, gels are dense aggregates of mature fibrils. A number of gelling peptides show characteristic cross-β structure reflexes using X-ray diffraction analysis [8,17]. This confirms their formation from fibrils.

The Phe-Phe and Ile-Phe dipeptides are the smallest peptides capable of assembling into fibrillar structures and forming microgels [11,18]. The Phe-Phe dipeptide corresponds to residues 19 and 20 of the central hydrophobic cluster of the amyloidogenic peptide Aβ(1–42) and has a strong effect on the Aβ peptide fibrillation [19]. The replacement of Phe by Ile preserves the ability of the dipeptide to form fibrils and microgels [11]. Interestingly, the Val-Phe dipeptide with Phe replaced by Val does not form fibrils [11].

The ability to form microgels has been detected in many small proteins and peptides [6,7,8,9,10,11,12,13,14,15,16,17,20]. Short peptide-based microgels are easy formed; they can change their properties depending on the external conditions of their formation (temperature, pH, ionic conditions, agitation, etc.) and modifications of the primary structure. Therefore, it is possible to create peptides that form gels with desired properties by changing the pH or introducing various modifications into the sequence [9,17].

The possibility to regulate the formation of gels by changing the pH is important in pharmacology. Microgels can be used as a matrix for drug delivery to various parts of the digestive tract. Their swelling and release of medicinal components will occur in certain parts of it: either in the stomach (pH 1–2) or in the intestine (~pH 7.0). Various polymer gels with variable properties are used as material for drugs shells (derivatives of polyvinyl alcohol, cellulose, starch, acrylamide).

However, the main advantage of gels based on natural short proteins and peptides is their biocompatibility and degradability. In the work [15], it was shown that a small protein, lysozyme, forms fibrillar structures, and on its basis, gel shells (coatings) for the inclusion of such well-known antibiotics as penicillin and tetracycline were obtained. 

Short peptides are convenient for studying the mechanism of fibril formation. They are easily synthesized and are suitable for studying the effect of various sequence modifications on the formation of fibrils, hydrogels and studying their properties [17,21]. In [17], for example, a seven-membered peptide (IKHLSVN) was used as a basis for sequence design in order to obtain its derivatives that easily form a gel. Small modifications of the side chain (Tyr, Phe) and the N-terminus of the peptide led to a change in the behavior of the peptides during amyloid formation and gel formation. All peptide modifications resulted in the formation of gels. The gels were investigated by EM, CD, X-ray diffraction analysis and ThT binding. The peptide modifications affected fibril morphology and X-ray diffraction data. Only one Phe-modified peptide had all characteristics of an amyloid peptide. Since the parent peptide self-assembles into fibrillar structures, but does not form a gel, the previously noted importance of aromatic groups for the process of amyloid formation and gel formation has been proved [17].

In addition, short peptides, which are amyloidogenic fragments of natural proteins/peptides with various modifications, are attractive building blocks for the formation of soft 3D materials used as a basis for the cultivation of various cell cultures, including stem cells. Based on the modified amyloidogenic fragment of the C-terminus of the Aβ(1–42) peptide, a number of biogels suitable for 2D/3D cell cultivation and stem cell differentiation were obtained [21]. It is interesting to note that these microgels showed non-toxicity and thixotropy, that is, the ability to restore the gel-like structure after mechanical destruction (shaking).

Current bioinformatics methods of amino acid sequence analysis allow predicting amyloidogenic fragments in the protein/peptide structure. In this work, we investigated the properties of amyloidogenic fragments capable of forming amyloid gels. Early studies of the Aβ(1–42) peptide associated with Alzheimer’s disease identified amino acid sequence regions that determine the amyloidogenic properties of the Aβ(1–42) peptide [22,23]. Two amyloidogenic fragments were identified in the Aβ(1–42) peptide. Both of these fragments are part of the 10-membered peptides synthesized by us (Aβ(16–25), Aβ(31–40) and Aβ(33–42)) and studied by various methods. Fragment Aβ(33–42) was investigated to clarify the role of the last two amino acids on the formation of fibrils [24]. Both fragments were found to form fibrils, exhibiting amyloidogenic properties (fibrillar structure, corresponding X-ray reflexes, ThT binding) according to EM, X-ray diffraction analysis and ThT data. 

We also studied 10-membered amyloidogenic fragments from the Bgl2p-glucantransferase of the *S. cerevisiae* cell wall [25,26]. It was shown that this protein is capable of forming fibrils under weakly acidic conditions. Three potential amyloidogenic determinants (PADs) were identified in the primary structure of this protein. These regions were included in 10-membered fragments, synthesized and investigated for the possibility of forming fibrils [25]. These amyloidogenic fragments were found to be sensitive to the pH of the medium. The first two fragments form fibrils in a wide range of pH (3.2–7.6), and the last one at pH 3.2–5.0. The last fragment and its modified homologue with the substitution of an amino acid in the second position, respectively, Asp to Glu, and both with blocked N-ends (AspNB and GluNB), were studied extensively to determine the effect of the substitution on the behavior of the peptide during fibrillation [26]. Both peptides form fibrils similar in morphology, and have characteristic reflexes in X-ray diffraction analysis for amyloids, but the replacement of Asp by Glu leads to a slowdown in the formation of mature fibrils from the GluNB peptide. 

Previously, studying the process of fibril formation by the Aβ(31–40), Aβ(33–42), AspNB and GluNB peptides, we found that fibrils are formed from ring-like oligomers, which interact with each other either ring to ring or ring on the ring with some shift [24,26]. On this basis, we assumed that a ring-like oligomer could be the building block of a fibril. Hence, a possible mechanism of the fibril formation has been suggested with the scheme: monomer → destabilized monomer → ring-like oligomer → fibril. Since the ring-like oligomers interact with each other in different ways, they form morphologically different fibrils.

In this work, we used EM and X-ray diffraction to determine the morphological and structural features of the short amyloidogenic peptides Aβ(31–40), Aβ(33–42), AspNB and GluNB, which, being concentrated, were able to form gels. According to the EM data the concentration of peptide preparations leads to the aggregation of fibrils, their compaction and disintegration of fibrils into shorter fragments up to their disassembly into building blocks—ring-like oligomeric structures. During the formation of fibrils, according to the data of fluorescence analysis, ThT binding occurs, and X-ray diffraction analysis indicates the presence of a cross-β structure in the samples.

Recently, gels of various natural peptides have been actively studied, since they are widely represented in organisms from humans to bacteria and are biocompatible. Moreover, the influence of various conditions of both the external environment and the amino acid sequence on the rate of fibril formation, their morphology and properties are shown. The ability to change the properties of fibrils and amyloid gels through modifications of their amino acid sequences and the creation of peptides with predetermined properties make them promising objects for the creation of various materials for pharmacology.

## 2. Results

### 2.1. Electron Microscopy Analysis of Gels of GluNB and AspNB Peptides

According to EM data, both peptides are capable of forming fibrils. At C = 0.5 mg/mL under conditions of 5% acetic acid (pH 3), 5% DMSO, after 8 h of incubation at 37 °C, mature fibrils up to several microns in length and the thinnest diameter of about 8 nm are observed. With an increase in the incubation time to 26 h, the number of mature fibrils increases, and their length reaches 10–12 microns (Figure 1). With an increase in the incubation time to 48 h, a barely noticeable transparent gel-like precipitate begins to appear on the walls of the tubes. At the same time, according to EM data, the number of fibrils in the solution sharply decreases, which indirectly indicates that most of the fibrils turn into a gel-like precipitate. The EM images show that such a gel-like precipitate is formed by dense aggregates (clusters) of intertwined mature fibrils (Figure 1).

With an increase in the concentration of peptide preparations to 1 mg/mL, the formation of a gel-like precipitate occurs much faster: for GluNB after 24 h, for AspNB after 8 h of incubation. At C = 2 mg/mL, gel formation occurs after 1–2 h of incubation. After centrifugation of preparations with C = 0.5–1.0 mg/mL at 12,000 rpm (30 min) and removal of the supernatant, the gel is clearly visible in test tubes (Figure 2).

EM analysis of the gel showed that it consists of large aggregates (clusters) of fibrils. A lower concentration of fibrils can be observed at the edges of the clusters. This allows us to draw some conclusions about the structure of the gel based on the morphological characteristics of the preparation (Figure 3). A large number of short (up to 50 nm) fragments of mature fibrils and amorphous aggregates of various sizes are clearly observed. At higher magnification, it can be seen that short fibrils and small aggregates consist of ring-like oligomers with an average diameter of about 7–8 nm (Figure 3). We observed the same ring oligomers earlier when studying the process of fibrillation of peptides GluNB and AspNB [26]. Based on the EM data, it can be concluded that at a sufficient concentration of the preparation, the process of gel formation occurs through the aggregation of long (up to several μm) mature fibrils into large dense clusters. The aggregation of fibrils into clusters is related to the fragmentation of fibrils into small fragments. With further compaction of the fibril fragments, the formation of a gel occurs. The structure of the gel can be determined by the amorphous aggregates in the preparation. With sufficient magnification, it can be seen that amorphous aggregates, likely short fibrils, are formed from ring-like oligomers (Figure 3). Ring-like oligomers in the fibril are located either oligomer to oligomer or slightly overlap each other, which is consistent with our data obtained earlier in the study of the mechanism of GluNB and AspNB peptide fibrillation [26].

### 2.2. Electron Microscopy Analysis of Gels of Aβ(31–40) and Aβ(33–42) Peptides

Both amyloidogenic fragments of the Aβ(1–42) peptide are capable of forming gels. According to EM data, after 6–8 h of incubation of a peptide with a concentration of 0.5 mg/mL under conditions of 5% DMSO, 50 mM Tris-HCl, pH 7.5 and 37 °C, mature fibrils up to several microns in length and a thinnest diameter of about 8 nm are observed. With an increase in the incubation time to 26 h, fragments of gels appear on the walls of the tubes. According to early studies, the morphology of fibrils of peptides Aβ(31–40) and Aβ(33–42) differs significantly [24]. Fibrils of the Aβ(31–40) peptide have a tendency for lateral association, which leads to the formation of associates in the form of wide ribbons or bundles (Figure 4).

Interestingly, the Aβ(1–40) peptide also has a tendency to form ribbons under similar conditions [27]. Mature fibrils of the Aβ(33–42) peptide are significantly heterogeneous in diameter (8–35 nm) and have a rough surface, which is also characteristic of the Aβ(1–42) peptide fibrils [27]. After 8 h of incubation, both peptides begin to aggregate with the formation of a gel visible on the walls of the tubes (Figure 5).

After 26 h of incubation and centrifugation at 12,000 rpm (30 min), a gel appears at the bottom of the tubes, as in the study of peptides GluNB and AspNB. According to EM data, the gel of both preparations is formed by dense fibril aggregates. At the edges of the gel, it can be seen that its constituent fibrils are noticeably shorter than mature fibrils formed at earlier stages, and reach a length of 40 nm for Aβ(31–40) and 80–100 nm for Aβ(33–42). It should be noted that the transition of fibril preparations to a gel-like state occurs faster for Aβ(31–40) peptide than for Aβ(33–42) (a gel-like precipitate appears in the test tube by 6 h of incubation). At higher magnification, it can be seen that short fibrils in the gel are formed, like mature fibrils of peptides Aβ(1–40) and Aβ(1–42) [22,23], from ring-like oligomers that interact with each other in different ways. Ring-like oligomers of Aβ(31–40) are mainly localized either ring to ring or slightly overlap each other. The same association of ring-like oligomers is also characteristic of the parent Aβ(1–40) peptide. The ring-like oligomers of the Aβ(33–42) peptide interact with each other in an irregular manner and, as a result, the mature fibrils acquire a rough surface and differ significantly in diameter, which resembles the morphology of the fibrils of the Aβ(1–42) peptide [28].

Thus, as in the case of peptides GluNB and AspNB, the gels of the Aβ(31–40) and Aβ(33–42) peptides consist of short fragments of mature fibrils, which are formed by ring-like oligomeric structures with a diameter of about 7–8 nm (Figure 6).

It should be noted that, along with short fibrils, EM analysis of gel preparations reveals accumulations of single ring-like oligomers. That is, during the formation of the gel, fragmentation of fibrils into the oligomeric state can occur.

### 2.3. X-ray Diffraction Analysis

One of the important characteristics of amyloid fibrils is the presence of a cross-β structure, which is revealed by X-ray diffraction analysis. According to X-ray diffraction analysis data, preparations of gel-like sediments of amyloidogenic fragments GluNB, AspNB, Aβ(31–40) and Aβ(33–42) show two specific reflexes characteristic of cross-β structural organization: meridional 4.5–4.9 Å and equatorial 8–12 Å. Figure 7 shows the diffraction patterns of preparations of gels of the GluNB and Aβ(31–40) peptides. The preparations of gels of AspNB and Aβ(33–42) give similar patterns.

## 3. Discussion

According to a simplified scheme, the formation of fibrils begins with the destabilization of the native protein/peptide molecule, then such altered monomers are associated with the formation of oligomers, and, finally, the association of oligomers leads to the formation of fibrils. The stage of transition of proteins/peptides from the oligomeric state to the fibrillar one remained an obscure moment in this process. We have previously shown that destabilized monomers of various short proteins/peptides form ring-like oligomeric structures, which can be the main building blocks for the formation of fibrils [22,24]. These ring oligomers interact with each other in different ways depending on the amino acid composition, and this affects the morphology of mature fibrils [24,26,28]. We carried out a study of the gels’ morphology and found that gels are dense aggregates of fibrils, which is consistent with many previous studies [7]. However, most authors used a high dilution of protein/peptide gels for EM analysis. Additionally, dense networks of gels were observed using AFM or scanning electron microscopy. These methods may not be suitable for observing the fine structure of gels. We made an attempt to study gels with TEM and used preparations with minimal dilutions. This allowed us to observe the change in the morphology of fibrils during the formation of gels and to analyze their morphology. The data obtained allow us to conclude that the process of gel formation can be described as follows: with a sufficient concentration of proteins/peptides, mature fibrils interact with each other, forming large fibrillar clusters; in clusters, fibrils are compacted, which results in their fragmentation into shorter fragments up to oligomers; further compaction of short fragments of fibrils and ring-like oligomers leads to a change in the structure of aggregates and to the transition of preparations into a gel-like amorphous state. Thus, the gel is a very dense association of short fibrils and individual (single) ring-like oligomers. According to the gel formation scheme, oligomers are the initial building blocks for fibril formation, as well as the final products during gel formation (Figure 8).

It should be especially noted that in the process of gel formation, short fragments of fibrils and oligomers create a dense network that can prevent the normal diffusion of various substances both into the cell and out of it, which affects the normal metabolism of cells, up to their death.

X-ray analysis indicates the presence of cross-β structure in the gel preparations of all analyzed peptides, since reflexes characteristic of amyloids are observed (4.5–4.9 Å and 8.0–12 Å). However, according to EM data, the gel is formed by aggregates of short fibril fragments (20–40 nm) and ring-like oligomers. This indicates that short fibril fragments already contain a cross-β structure. Thus, the presence of a cross-β structure cannot be an unambiguous proof that the β-sheets are formed from β-strands located along the entire fibril, since the presence of short β-sheets in the structure of fibril fragments is sufficient for the formation of reflexes with values of 4.5–4.9 and 8.0–12 Å.

Earlier, we showed that, depending on the amino acid composition, the morphology of fibrils can differ. In the Aβ(1–40) and Aβ(1–42) peptides, this relationship is especially pronounced. Many authors have noticed that Aβ(1–40) fibrils are smoother and laterally well associated, easily forming ribbons and bundles. The Aβ(1–42) peptide is characterized by greater polymorphism. It forms irregular rough fibrils that form bundles of different diameters [22]. Interestingly, the same uneven fibrils are characteristic of the amyloidogenic fragment Aβ(33–42). Our early EM data showed that the studied peptides are composed of ring-like oligomers. Analyzing EM images of amyloidogenic proteins and peptides made by other authors, one can notice a similar organization of fibrils from ring-like oligomers [29,30]. In a number of recent works, one of the possible mechanisms of fibril formation is considered the process of the interaction of oligomer/granules with each other [24,28,31,32]. The organization of fibrils through the interaction of ring-like oligomers with each other in different ways makes it possible to explain such properties of fibrils as polymorphism, fragmentation and branching. In our opinion, X-ray diffraction data provide additional evidence that the main building blocks of fibrils are ring-like oligomers. Such ring-like oligomers can be organized at the molecular level in different ways depending on the amino acid composition, but the general principle of fibril organization (association of ring-like oligomers) is preserved [33].

## 4. Materials and Methods

### 4.1. Chemical Synthesis of Amyloidogenic Peptides: Aβ(31–40), Aβ(33–42), AspNB and GluNB

The reagents employed in the preparation of the peptides were supplied by Fluka, IRIS Biotech GMBH (KHIMMED, Moscow, Russia) and were used as received. Samples of peptides Aβ(31–40) (_31_IIGLMVGGVV_40_) and Aβ(33–42) (_33_GLMVGGVVIA_42_) were prepared starting with Wang resin (0.5 g, 0.26 mmol/g initial loading). The details of the solid phase peptide synthesis methodology employed in the preparation are described elsewhere [34,35,36] and these common protocols were followed throughout the synthesis. Whenever peptidyl polymer aggregation occurred, coupling protocols were modified either by employing various combinations of polar aprotic solvents instead of dimethyl formamide (DMF) or by using so-called “chaotropic mixtures”, such as potassium thiocyanate (4M KSCN) in DMF [36]. The crude peptide was released from the polymer with concomitant removal of the acid-labile protecting groups using standard TFA-TIPS-H_2_O mixture of 95:2.5:2.5 (*v*/*v*/*v*) at room temperature for 90 min. The crude peptides were purified by preparative HPLC and the collected fractions were analyzed by an Orbitrap Elite mass spectrometer (Thermo Scientific, Dreieich, Germany). Appropriate fractions were combined and lyophilized providing target peptides as white amorphous non-hygroscopic solids. Aβ(31–40): yield 19 mg (ca. 15%), for [M + H^+^] C_44_H_81_N_10_O_11_S found 957.5784, calculated 957.5802, ∆M = 2 ppm. Aβ(33–42): yield 17 mg (ca. 14%), for [M + H^+^] C_41_H_75_N_10_O_11_S found 915.5314, calculated 915.5332, ∆M = 3 ppm. The preparation of peptides AspNB and GluNB was reported earlier [26].

### 4.2. Samples

To obtain the monomeric state of synthetic peptides GluNB, AspNB, Aβ(31–40), Aβ(33–42), preparations were dissolved in NH_4_OH, the concentration was measured and aliquots with the required concentration (0.2–1.0 mg/mL) were prepared and lyophilized. Prior to the experiment, aliquots were dissolved in 100% DMSO, then adjusted to the desired concentration with 5% acetic acid (pH 3) for GluNB and AspNB and 50 mM TrisHCl (pH 7.5) for Aβ(31–40) and Aβ(33–42). The final concentration of DMSO in the preparations was about 5%. Dissolution of preparations was carried out at 4 °C, and incubation at 37 °C.

### 4.3. Electron Microscopy

At certain time intervals, aliquots of the preparations were taken for EM analysis and prepared by the negative staining method. Before staining, all preparations were adjusted to a concentration of 0.2 mg/mL with the appropriate buffers: GluNB and AspNB with 5% acetic acid, Aβ(31–40) and Aβ(33–42) with 50 mM Tris-HCl (pH 7.5). A copper grid (400 mesh) coated with a formvar film (0.2%) was mounted on a sample drop (10 µL). After 5 min of absorption, the grid was negatively stained for 1.5–2.0 min with 1% (weight/volume) aqueous solution of uranyl acetate. The excess of the staining agent was removed with filter paper. Gel-like precipitates for all the preparations begin to form at C = 0.5–1.0 mg/mL after 8–16 h of incubation at 37 °C, and fragments of the gels were clearly visible on the walls of the tubes.

Gel preparations for EM analysis were prepared as follows. A small gel fragment (about 1 mm in diameter) was taken with a pipette and carefully dissolved (mixed by pipetting) in 20 μL of the corresponding buffer. About 5 μL of the preparation was applied to the copper grids coated with a formvar film. After adsorption of the preparation (1–2 min), the mesh was washed on a drop of buffer for 30 s. Contrasting with uranyl acetate preparation was carried out as described above. The preparations were analyzed using a JEM-100C and JEM1200EX (JEOL, Tokyo, Japan) transmission electron microscope at an accelerating voltage of 80 kV. Images were recorded onto Kodak electron image film (SO-163) (Kodak Electron Image Film, New York, NY, USA) at a nominal magnification of 40,000.

### 4.4. X-ray Diffraction Analysis

For X-ray analysis, the samples of peptides were prepared after 24–48 h of incubation in appropriate buffers at 37 °C at the concentration of 0.5–1.0 mg/mL: GluNB and AspNB under conditions of 5% acetic acid (pH 3), 5% DMSO; Aβ(31–40) and Aβ(33–42) under conditions of 50 mM TrisHCl (pH 7.5), 5% DMSO. The samples were centrifuged at 12 rpm for 10 min at room temperature to precipitate large aggregates; the supernatant was removed and a drop of a thick gel-like precipitate of each peptide (6–7 μL) was placed in the space (about 1.0 mm) between the tips of two glass rods with a diameter of about 1 mm, coated with wax. Then, the preparations were dried for several hours in a Petri dish. The fiber diffraction images were collected using a Microstar X-ray generator with HELIOX optics, equipped with a Platinum135 CCD detector X8 Proteum System (“Bruker AXS”, Karlsruhe, Germany) at the Institute of Protein Research, RAS, Pushchino. Cu Kα radiation (λ = 1.54 Å) was used. The samples were positioned at a right angle to the X-ray beam using a 4-axis kappa goniometer (“HUBER”, Rimsting, Germany).

## 5. Conclusions

We studied the morphology of the gels and found that the gels are dense fibril aggregates. We tried to study the gels using TEM and used preparations with minimal dilutions. This allowed us to observe the change in the morphology of fibrils during the formation of gels and to analyze their morphology. It turned out that the gel is a very dense association of short fibrils and separate (single) oligomers. According to the gelation scheme, oligomers are the initial building blocks for fibril formation as well as the final products in the gelation process.

In our opinion, such a mechanism of fibril assembly indicates that the main attention in the development of drugs for the prevention and treatment of various types of amyloidosis should be paid to the initial stages of fibril formation. To prevent the formation of oligomeric structures toxic to cells, it is necessary to establish the factors leading to changes in the structural organization of proteins/peptides and find ways to eliminate them. This means that in the development of drugs, attention should mainly be paid not to the destruction of fibrils, but to the prevention of ring-like oligomeric complex formation [33]. This strategy is becoming more and more relevant as more and more data confirm the importance of biogels and biofilms for the normal functioning of organisms. If, for bacteria, biofilms are an important protective formation that promotes the colonization of an infected organism (tubercle bacillus and other pathogens), then in higher organisms, various stress factors can lead to the formation of extracellular gels that prevent the diffusion of substances necessary for normal cell metabolism and can lead to serious consequences up to cell death.

## Figures and Tables

**Figure 1 nanomaterials-11-03129-f001:**
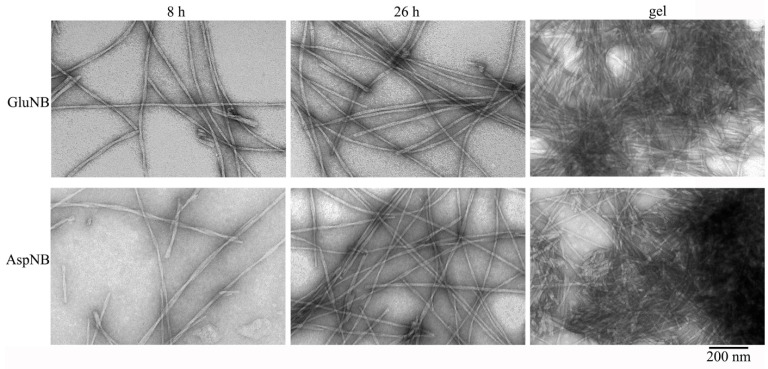
EM images of GluNB and AspNB peptide preparations. All experiments were performed under conditions of 5% DMSO, 5% acetic acid (pH 3.0), concentration 0.5 mg/mL, incubation at 37 °C.

**Figure 2 nanomaterials-11-03129-f002:**
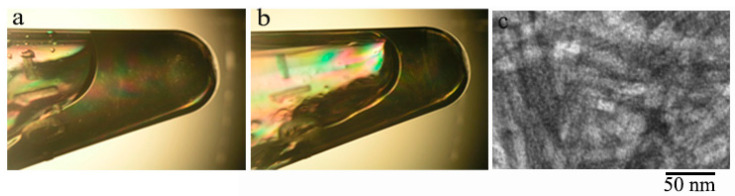
Gel-like precipitate formed by: (**a**) GluNB peptide and (**b**) AspNB peptide; (**c**) EM image of the GluNB gel preparation. Preparations under conditions of 5% DMSO, 5% acetic acid (pH 3.0), concentration 0.5 mg/mL and incubation at 37 °C for 48 h.

**Figure 3 nanomaterials-11-03129-f003:**
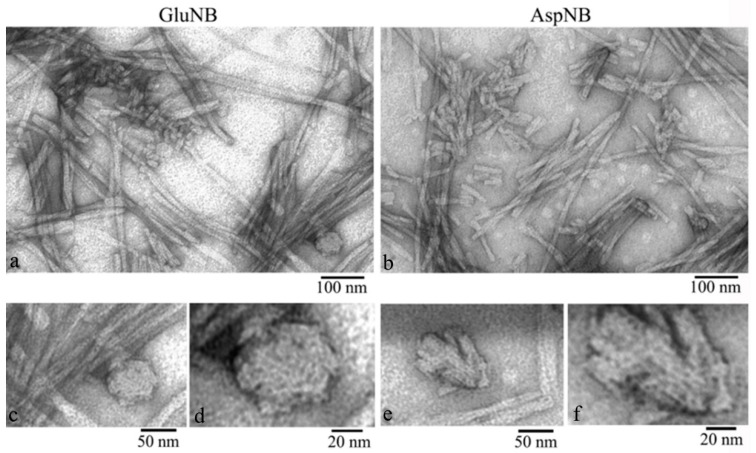
EM images of the edge regions of the GluNB and AspNB peptide gels. Fields of samples: (**a**) GluNB; (**b**) AspNB. Fragments of fields at higher magnification: (**c**,**d**) GluNB; (**e**,**f**) AspNB.

**Figure 4 nanomaterials-11-03129-f004:**
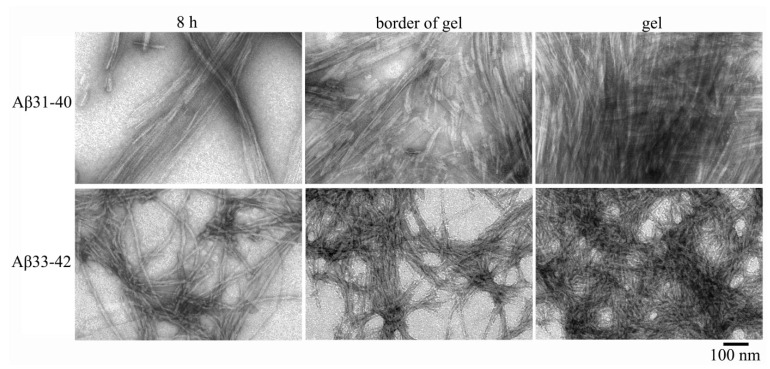
EM images of the peptides Aβ(31–40) and Aβ(33–42). The preparations were made at 0.5 mg/mL under conditions of 5% DMSO, 50 mM Tris-HCl, pH 7.5 at 37 °C.

**Figure 5 nanomaterials-11-03129-f005:**
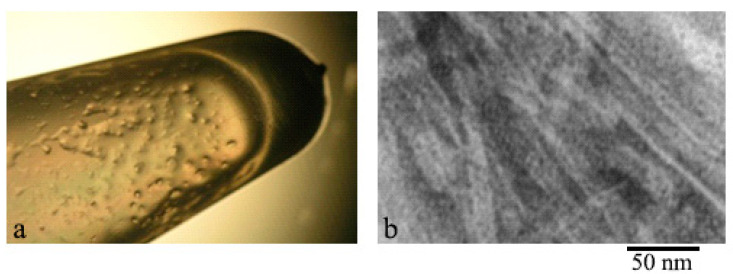
Gel-like precipitate formed by the Aβ(31–40) peptide: (**a**) gel on the walls of the test tube; (**b**) EM image of a gel preparation at 0.5 mg/mL under conditions of 5% DMSO, 50 mM Tris-HCl, pH 7.5 at 37 °C after 26 h of incubation.

**Figure 6 nanomaterials-11-03129-f006:**
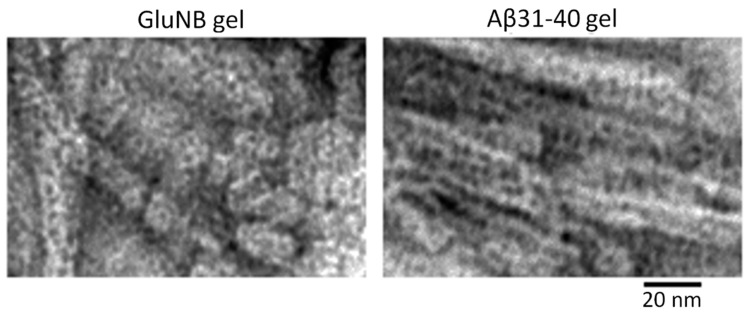
EM images of preparations of GluNB and Aβ(31–40) peptide gels. Short fibrils and ring-like oligomeric complexes are present.

**Figure 7 nanomaterials-11-03129-f007:**
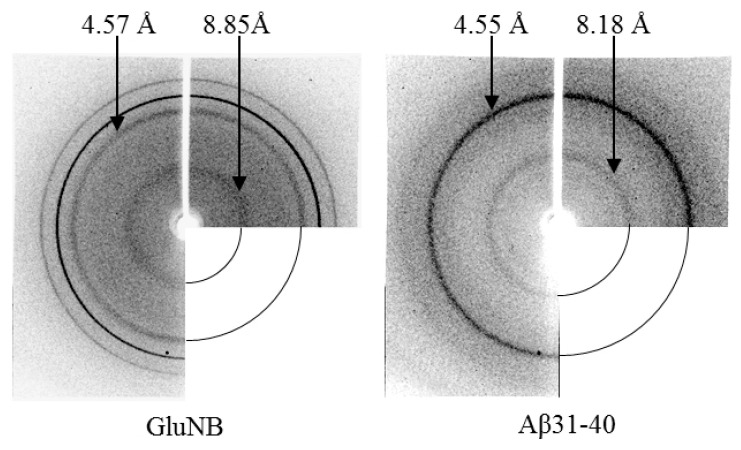
X-ray diffraction patterns of gels of GluNB and Aβ(31–40) peptides.

**Figure 8 nanomaterials-11-03129-f008:**
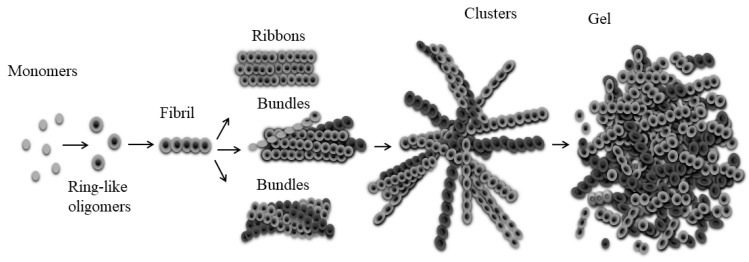
Scheme demonstrating the mechanism of gel formation.

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
