# Peer review of "Mechanism of Amyloid Gel Formation by Several Short Amyloidogenic Peptides"

_nanomaterials, 2021, doi:10.3390/nano11113129_

Round 1

Reviewer 1 Report

The manuscript titled, "Mechanism of Biogel formation" by Galzitskaya, O.V., explains the biogel formation by GluNB and AspNB peptides inspired from S.cerevisiae cell wall. The work is interesting and please read on for my comments.

  1. The Introduction is fairly long and can be shortened.
  2. This manuscript reports the same work presented in 10.1016/j.bbapap.2016.08.002 with inclusion of XRD data in place of MD data. This does not essentially explain the mechanism of biogel formation. 
  3. All of the data presented in the current manuscript explains that the peptides GluNB and ASPNB forms biogel at certain concentrations under appropriate conditions. But the mechanism is not explained clearly anywhere.
  4. The difference in the mechanism of gel formation by GluNB and AspNB is never been established or discussed.
  5. What is the comparison of significance between AB(31-30) and AB(33-42) and with that of gels from GluNB and AspNB.
  6. Overall, the manuscript can be improved with additional data either by generating mutants or specify the difference and significance of Glu or Asp residues in GluNB/AspNB.

Reviewer 2 Report

In general, I think the manuscript entitled “Mechanism of Biogel Formation” can be published. Yet, I have the following comments for the authors to consider.

  • The title is so broad. I think it can be mechanism of biogel formation by several short amyloidogenic peptides.
  • I think using amyloid gel is better than using biogel.
  • One unique feature of this work is that the author proposed a new mechanism of amyloid formation based on ring-like oligomer structure. This model is completely different from the common amyloid formation model. I think the authors should put more discussion on this model in this paper as most people in the field are not familiar with this model.
  • In the TEM images, the authors should circle those ring structures clearly.
  • In figure 8, there are two types of bundles. Please provide more discussion on the difference between the two models.

Author Response

In general, I think the manuscript entitled “Mechanism of Biogel Formation” can be published. Yet, I have the following comments for the authors to consider.

1) The title is so broad. I think it can be mechanism of biogel formation by several short amyloidogenic peptides.

Answer:

Thank you for your suggestion. We have changed.

2) I think using amyloid gel is better than using biogel.

Answer:

We have changed.

3) One unique feature of this work is that the author proposed a new mechanism of amyloid formation based on ring-like oligomer structure. This model is completely different from the common amyloid formation model. I think the authors should put more discussion on this model in this paper as most people in the field are not familiar with this model.

Answer:

We write about amyloid formation of these peptides in the Introduction section and we have published many papers about our model.

“Previously, studying the process of fibril formation by peptides Aβ(31-40), Aβ(33-42), AspNB and GluNB, we found that fibrils are formed from ring-like oligomers, which interact with each other either ring-to-ring or ring on the ring with some shift [24,26]. On this basis, we assumed that a round-like oligomer could be the building block of a fibril. Hence, it has been suggested a possible mechanism of the fibril formation by the scheme: monomer – destabilized monomer – ring-like oligomer – fibril. Since the ring-like oligomers interact with each other in different ways, they form morphologically different fibrils.”

4) In the TEM images, the authors should circle those ring structures clearly.

Answer:

Figure 3 (Fragments of fields at higher magnification: (c, d) GluNB; (e, f) AspNB.) and Figure 6 (EM images of preparations of GluNB and Aβ(31-40) peptide gels) present short fibrils and ring-like oligomeric complexes.

5) In figure 8, there are two types of bundles. Please provide more discussion on the difference between the two models.

Answer:

Any long polymer is twisted, including fibrils. We presented two types of bundles, where they are intertwined and not.

Round 2

Reviewer 1 Report

The reviewer report has adequate answers and hence I recommend publication!